# Survival of the fittest in the pandemic age: Introducing disease-related social Darwinism

**Paul Nachtwey** [ID] [¤] [�***], **Eva Walther** * [�***]

Department of Psychology, University of Trier, Trier, Rheinland-Pfalz, Germany

***  These authors contributed equally to this work.
¤  Current address: Faculty of Behavioural and Movement Science, Department of Social Psychology, Vrije Universiteit Amsterdam, Amsterdam, Netherlands
* walther@uni-trier.de

**Data Availability Statement:** All Data, SPSS and R Syntax, the preregistration for Study 2 and the online supplemental materials are available on OSF: DOI 10.17605/OSF.IO/28XPG.

## Abstract

COVID-19 was a harsh reminder that diseases are an aspect of human existence and mortality. It was also a live experiment in the formation and alteration of disease-related attitudes. Not only are these attitudes relevant to an individual's self-protective behavior, but they also seem to be associated with social and political attitudes more broadly. One of these attitudes is Social Darwinism, which holds that a pandemic benefits society by enabling nature "to weed out the weak". In two countries ($N = 300$, $N = 533$), we introduce and provide evidence for the reliability, validity, and usefulness of the Disease-Related Social Darwinism (DRSD) Short Scale measuring this concept. Results indicate that DRSD is meaningful related to other central political attitudes like Social Dominance Orientation, Authoritarianism and neoliberalism. Importantly, the scale significantly predicted people's protective behavior during the Pandemic over and above general social Darwinism. Moreover, it significantly predicted conservative attitudes, even after controlling for Social Dominance Orientation.

## Introduction

The COVID-19 pandemic has not only painfully reminded us that diseases are part of a human's life and death, but was also a real-time experiment in disease-related attitude formation and change. These attitudes are not only relevant to an individual's behavior but were a source of great political polarization in different countries around the world. This polarization was, for example, evident in politicians' communication [1]. While some politicians around the world appealed to people's sense of community and solidarity in order to fight the pandemic, others seemed to prefer to let the virus run its course in hopes of herd immunity and better economic outcomes. The Lt. Governor of Texas, for example, suggested that the elderly should risk their lives in order not to ruin America [2]. That is, he implied a willingness to sacrifice the "weaker" people for an alleged benefit to society. In other words, he seemed to refer to an attitude relatively unnoticed in Social Psychology so far: Social Darwinism.

### Social Darwinism

Social Darwinists apply Darwin's biological theory of natural selection to the evolution of human society. Hence, society is perceived as an organism that evolves through the process of

**Funding:** The authors received no specific funding for this work.

**Competing interests:** The authors have declared that no competing interests exist.

natural selection [3]. A central idea of social Darwinism is expressed by the term "survival of the fittest", introduced by Herbert Spencer as early as 1852 [4]. Accordingly, "for the good of the species, nature should be allowed to weed out the weak" [5 p1141], in other words: the struggle for resources should be maximized as a way of getting rid of the weaker members of society, which allows the naturally strong to thrive [6]. A key element of this thinking is that natural selection equals progress. Consequently, social Darwinists reject governmental interventions intended to strengthen the weak or to create social equality, as such attempts are seen as a threat to societal progress [7]. Hence, inequality is seen as both, inevitable and preferable for people holding social Darwinist attitudes [5].

Social Darwinism has influenced politics ever since the idea was first popularized by Spencer. Martin [8], for example, argued that social Darwinism was partly responsible for the justification of a laissez-faire approach to charity and social welfare throughout the 19th and 20th century in the US. Moreover, both Leyva [7] and Tienken [9] stated that certain educational reforms in the US (the No Child Left Behind Act) have been influenced by social Darwinist ideas packaged as neoliberalism. Leyva [7] extended this argument, stating that social Darwinism has constantly resurfaced in neoliberal economics and free-market policies in the US.

**Social Darwinism and COVID-19.** Thus, although social Darwinism has been widespread in societies for a long time, it can be assumed that the COVID-19 pandemic has increased the prevalence of this thinking. As Strobl [10] pointed out, the extreme right in Austria and Germany used social Darwinist arguments to popularize their ideology in the context of the pandemic. In a study of German anti-vaxxers and anti-maskers, more than 60% showed at least partial agreement with social Darwinist ideas [11]. Analyzing samples from Turkey and the US Kanık et al. [12] investigated the role of social Darwinism and ageism in predicting opposition to policies designed to protect elderly people during the pandemic. They found that social Darwinism predicts lower support for such policies through predicting higher levels of ageism.

Considering that pandemics, like COVID-19, have become increasingly likely over the last years [13,14], it is a vital question how pandemics change and form political attitudes. Even though social Darwinist ideas about diseases seem to have revitalized during the current pandemic, no measurement exists so far that directly assesses disease-related social Darwinism; a specification of general social Darwinist attitudes applied to the realm of diseases. To address this issue, we developed the Disease-Related Social Darwinism (DRSD) Scale, to investigate how a social Darwinist view of diseases relates to people's health-related and political attitudes and behavior.

## Disease-related social Darwinism

As the COVID-19 pandemic has illustrated, diseases are part of a human's life and death and if they are not properly controlled, they can change the way we live together in a society. However, this is not only the case for diseases like COVID-19 but also for other infectious diseases like for example measles, if not enough children are vaccinated at the right time [15–17]. This highlights the importance of people's disease-related attitudes, not only for the individual's health behavior but for collective behavior as well. An attitude that seems to have been widespread during the pandemic is DRSD as it frames diseases (especially pandemics) as having a positive function for society.

**(Disease-related) social Darwinism and political orientation.** During the pandemic differences in social distancing and other disease-related behaviors between groups became evident. Often it seemed, that people on the right of the political spectrum were less concerned about COVID-19 or even outright angry at governments restricting public life to stop its

spread [18]. Using smartphone data, Alcott et al. [19], for example, showed that areas in the US with predominantly Republican voters engaged in less social distancing. This effect stayed significant even when controlling for other factors like public policies or population density. Republicans and Democrats also differed in their beliefs about the pandemic. Democrats perceived COVID-19 as more severe and social distancing as more effective than Republicans. Furthermore, majority Republican counties showed higher numbers of additional deaths due to COVID-19 than majority Democratic counties [20], this association also seems to hold on the individual level [21]. We argue that one attitude possibly explaining these differences is DRSD.

As pointed out earlier, a German study of anti-maskers and anti-vaxxers found high degrees of at least partial support for certain social Darwinist ideas [11]. The same study also found that these people were inclined to vote for the right-wing populist party "Alternative für Deutschland" (AfD; [22]) in the 2021 election. In a large representative German sample, social Darwinism was highly correlated with support for a right-wing authoritarian dictatorship [23]. And in a similar vein, a more recent investigation in Germany found that social Darwinism was most strongly endorsed by people identifying as right-wing and by people preferring the AfD [24]. It seems then that social Darwinism is an attitude more strongly pronounced on the right-wing end of the political spectrum, which should also be the case for DRSD as it refers to a specification of social Darwinism.

**(Disease-related) social Darwinism and social attitudes.** Previous research on social Darwinism has revolved mainly around a) investigating it as part of an extreme right-wing ideology in Germany [23] or b) investigating naïve social Darwinism/Competitive Jungle Beliefs in light of a dual-process model of motivation [25]. However, two recent studies by Saud [26] and Rudman and Saud [5] conceptualized social Darwinism as a central inequality justifying social attitude. The authors defined social Darwinism as the belief "that humans, like plants and animals, are engaged in a ruthless genetic competition" [5 p1141]. Utilizing this concept, Saud [26] found that social Darwinism, was related to prejudice against minorities, Social Dominance Orientation (SDO; [27]), and Right-Wing Authoritarianism (RWA;[28]). That social Darwinism is related to SDO is not surprising, because it refers to the degree to which an individual prefers that his or her in-group is dominant over other groups in society [27]. Similarly, RWA should be related because it refers to the acceptance of hierarchies, the submission to authorities who uphold them, and aggression against those who oppose them [28]. These associations provide the basis for our understanding of DRSD, as a specific form of social Darwinism, in the context of social and political attitudes. Importantly, Rudman and Saud [5] have shown that social Darwinism explains more variance of three forms of system justification (gender system, race system, economic system) than SDO and biological essentialism [29]. The authors suggested that social Darwinism explains more variance because it provides both a rational explanation for and positive evaluation of social hierarchies, while SDO and biological essentialism provide only one or the other. Rudman and Saud [5] concluded that when individuals believe that differences between people are the result of natural selection, attempts to create equality may be rejected as a threat to human welfare and progress, because, according to this view, inequality as a result of natural selection is both inevitable and preferable.

**The role of DRSD during COVID-19.** Based on this logic, people with disease-related social Darwinist attitudes should perceive restrictions imposed by the government to stop the spread of COVID-19 as unnecessary or even dangerous because they could be seen as "attempts to level the playing field" [5]. To investigate this hypothesis, we developed the DRSD scale and examined its usefulness in Study 1. In Study 2 (preregistered), we provide further validation and extend the findings to a different sample and broader political context. To our

knowledge, there is not yet a measurement directly capturing social Darwinist ideas about diseases. Due to the importance of disease-related attitudes outlined above, we aim to close this gap by introducing the DRSD scale.

### Ethics statement

The study was conducted following the 2016 American Psychological Association Ethical Principles of Psychologists and Code of Conduct [30]. As the project did not involve deception, vulnerable populations, identifiable data, intensive data, or interventions, it was exempt from ethical approval at the participating institution.

## Study 1

The main goal of Study 1 was to develop the DRSD scale and as such to investigate its internal structure and validity. Regarding the validity of the scale, we expected the DRSD scale to be correlated with the social Darwinism scale [5] (Hypothesis 1a) because we theoretically consider DRSD a specification of the broader construct of social Darwinism applied to diseases. We also predicted that the DRSD scale would be related to SDO and RWA (Hypothesis 1b). DRSD should correlate with SDO, first; because social Darwinism can be seen as a hierarchy-enhancing myth, legitimizing inequality and hierarchies in society. Second, as Rudman and Saud [5] pointed out, SDO and social Darwinism share conceptual roots in biological determinism, which makes them "ideological cousins" (p1141). The DRSD scale should further correlate with authoritarianism because the concept of social Darwinism justifies the acceptance of authoritarian figures. It was further hypothesized that the DRSD scale should be related to neoliberal thinking (Hypothesis 1c). The ideal of an unregulated market is a cornerstone of this ideology [31,32] and has been deeply entwined with social Darwinism throughout history [33–35]. Moreover, Azevedo et al. [32] have shown that the endorsement of neoliberal ideology correlates with SDO, which is, as already pointed out, deeply connected to social Darwinism. Furthermore, neoliberalism [32] and social Darwinism [5] can both be seen as justifications for inequality in society.

We also expect that political left-right orientation would predict DRSD (Hypothesis 2) because it has been shown that social Darwinism overlaps with (extreme) right-wing ideologies [23,24,36].

More importantly, however, we also want to show that the scale has incremental validity in comparison to the standard social Darwinism scale when predicting peoples' protective behavior during the COVID-19 pandemic (Hypothesis 3a). Furthermore, we think that the relationship between party preference and behavior during the pandemic will be moderated by DRSD (Hypothesis 3b). For an overview of our hypotheses see Fig 1.

### Method and materials

**Participants.** This study was conducted as an online study in German. Participants ($N$ = 304) were recruited via several methods. University student groups (some with a political agenda) were contacted via e-mail. Of the 28 contacted groups, 8 agreed to share the link to the survey with their respective groups. The link was also shared on an official e-mail platform of the University and social media (WhatsApp, Instagram, Facebook). We decided to recruit as close to the sample size of Rudman & Saud [5] as possible. We aimed for at least around 300 participants because it has been shown correlation coefficients stabilize around 250 participants [37], and a sample of this size is sufficient for factor analyses with less than 40 variables [38]. A post-hoc sensitivity analysis revealed that our final sample of 300 participants had an 80% power for detecting a minimum change in $R^2$ = .001 (when investigating on predictor

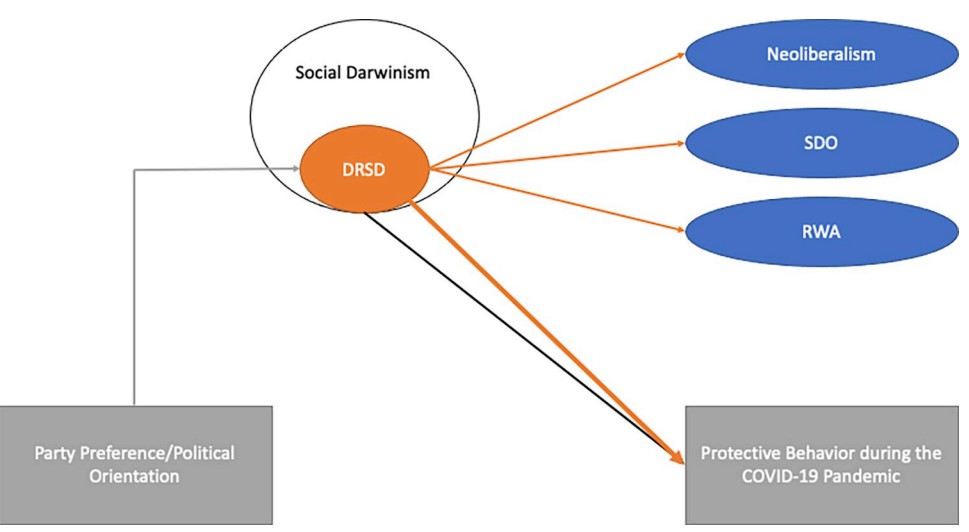

**Fig 1. Hypotheses Study 1.**

added to one or two other predictors). Based on pre-testing of our study we determined that the minimum time it takes to at least read all items would be around 5 minutes, therefore, four participants were excluded from the analysis as they took less than 5 minutes to complete the survey. The median time participants took to complete the survey was 13.63 minutes.

**Disease-related social Darwinism.** If not indicated otherwise, participants gave their response on a 7-point Likert scale ranging from "strongly disagree" (1) to "strongly agree" (7).

DRSD was assessed using a 24-item scale. Items were developed by the authors based on Rudman and Saud [5] and other sources [35,39,40]. An example of an item reads as: "Diseases are a natural mechanism to strengthen our immune system. It would be better for our future if we wouldn't try to prevent this" (for final scale and item statistics see online supplemental materials). The items were presented in a randomized order. Responses were averaged such that higher scores reflect stronger attitudes of DRSD. Because 5 of the final 14 items directly referred to rapidly changing official COVID-19 countermeasures, we dropped these items from further analysis. We used the remaining 9-item DRSD short scale for further analyses (for final short scale: $\alpha$ = .95, $M$ = 2.99, $SD$ = 1.63; more detailed descriptive statistics for relevant scales can be found in the online supplemental materials). Results, however, did not differ if the full 14-item scale was used. We believe that the high internal consistency of our scale provides evidence for the reliability of the scale.

**Social Darwinism.** We used a translation of the scale from Rudman and Saud [5]. The scale consists of 8 items (example item: "The fittest members of our society naturally rise to the top"). Responses were averaged such that higher scores reflected stronger endorsement of social Darwinist thinking ($\alpha$ = .88, $M$ = 2.86, $SD$ = 1.29).

**Social Dominance Orientation (SDO).** Preference for social hierarchy was measured using a German translation of the $SDO_{7(s)}$ scale from Ho et al. [41, Eckerle personal communication] (sample item: "an ideal society requires some groups to be at the top while others are at the bottom"). Responses to the 8 items were reversed coded when needed and then averaged such that higher scores indicated a stronger preference for inequality between groups in society ($\alpha$ = .83, $M$ = 2.56, $SD$ = 1.08).

**Right-Wing Authoritarianism (RWA).** RWA was assessed using the KSA-3 scale [42]. The KSA-3 is a German scale to measure authoritarianism based on the three subdimensions of authoritarianism proposed by, for example, Altemeyer [28]: authoritarian aggression

(general aggression against others sanctioned by authority figures), authoritarian submissiveness (submission to authority figures and general acceptance of their statements and actions) and conventionalism (strong obedience to established societal conventions). The scale has been proven useful in a German sample [42]. Responses were averaged such that higher scores indicated stronger authoritarian tendencies ($\alpha$ = .82, $M$ = 3.07, $SD$ = 1.04).

**Neoliberal ideology.** We measured neoliberal ideology using items of two subscales from Groß and Hövermann [43]: entrepreneurial universalism (orig: *unternehmerischer Universalismus;* a generalized version of neoliberal self-optimization) and competitive ideology (orig: *Wettbewerbsideologie;* the idea that in order to make progress in society, there needs to be omnipresent competition). The entrepreneurial universalism scale consists of three items (sample item: "If someone is not willing to try something new it's his own fault if he fails"), theescribtive ideology scale of two (sample item: "The key to success is to be better than everyone else"). Responses were averaged so that higher scores reflected a stronger endorsement of neoliberal ideology ($\alpha$ = .85, $M$ = 3.69, $SD$ = 1.37).

**Behavior and attitudes during the COVID-19 pandemic.** Two items were administered to measure whether participants took precautionary measures and followed official recommendations and guidelines during the COVID-19 pandemic. Participants were asked to indicate how strongly the two statements applied to themselves on a scale ranging from "does not apply at all" (1) to "applies completely" (7) (sample Item: "I follow the official guidelines regarding COVID-19"). Based on the high correlation ($r$ = .831, $p$ < .001), responses were averaged so that higher scores indicated more precaution and endorsement of official rules ($\alpha$ = .91, $M$ = 5.58, $SD$ = 1.42). We also administered one item to measure the attitudes of people regarding the official rules and policies concerning COVID-19. People were asked to indicate if they felt that the measures taken by the government were an overreaction on a scale ranging from "does not apply at all" (1) to "applies completely" (7) ($M$ = 3.60, $SD$ = 2.09).

**Political orientation.** Political orientation was measured using three items. Participants were asked to locate themselves on a scale from "very left" (1) to "very right (7) ($M$ = 3.26, $SD$ = 1.29), and they were also asked to rank themselves on a scale ranging from "very liberal" (1) to "very conservative" (7) ($M$ = 2.97, $SD$ = 1.28). Furthermore, we asked participants with which political party they could best identify (see Table 1). The parties were coded from most left (1) to most right (11); the options "other" and "none" were coded as (12) and (13). The left vs. right scale showed high correlation with party preference when people selecting "other" or "none" were excluded from analysis, justifying our classification of the parties ($r$ = .706, $p$ < .001, $N$ = 236).

**Table 1. Party Preference Study 1.**

| Party | N | Percent |
|---|---|---|
| Alliance 90/The Greens ("Bündnis 90 die Grünen", Green politics) | 82 | 27.3 |
| None | 49 | 16.3 |
| Christian Democratic Union of Germany ("CDU", liberal conservatism) | 42 | 14 |
| The Left ("Die Linke", democratic socialism) | 28 | 9.3 |
| Free Democratic Party ("FDP", classical liberalism) | 28 | 9.3 |
| Other | 15 | 5 |
| Social Democratic Party of Germany ("SPD", social democracy) | 14 | 4.7 |
| Alternative for Germany ("AFD", right-wing populism) | 12 | 4 |
| "Die Basis" (new party emerging from anti-vaxxer/masker communities) | 10 | 3.3 |

All other options were favored by less than 10 participants and are thus not mentioned. One person mentioned one party first and then "Die Basis" as a political group; this person was counted as a voter for the first party mentioned.

**Demographics.** Participants were asked to indicate if they were currently enrolled at a university; 51.0% (*N* = 153) were not. Furthermore, participants were asked to specify their gender (*N* = 162, 54.0% identified as female; *N* = 136, 45.3% as male, and *N* = 2 or .7% as diverse) and age (*M* = 37.73, *SD* = 18.72). We also asked participants to indicate whether their mother tongue was German, with 97.7% (*N* = 293) indicating that their mother tongue was German.

**Procedure.** After clicking on the link to the survey, participants were greeted and thanked for their participation. Following this, the measurements of DRSD, social Darwinism, SDO, RWA, and neoliberal ideology were presented in a randomized order on separate pages. At the end of the survey, participants' demographics were assessed. Afterward, participants were thanked and asked to share the survey with friends and family (for scales and instructions see online supplemental materials).

## Results

If not indicated otherwise, analyses for both studies were performed using SPSS version 26 [44].

**Factor analysis.** We conducted an exploratory factor analysis with oblimin rotation (varimax rotation yielded nearly identical results) of the 24 original items of the DRSD scale. Bartlett's test of sphericity was significant ($\chi^2$ (276) = 6061.40, $p < .001$), providing justification for the factor analysis. The scree-plot clearly suggested a one-factor solution (Fig 2). This factor explained 55.92% of variance. 14 of the 24 original items loaded highly only on this one factor (s. online supplemental materials for factor loadings). Because five items referred to rapidly changing anti-COVID measures, these items were excluded from further analyses, so that the final analyses rested on a 9-item short version of the scale. However, results did not differ significantly when the full scale was used. The analyses with the full 14-item scale are reported in the online supplemental material.

We also calculated an additional exploratory factor analysis on the social Darwinism scale and the 9-item DRSD-scale to investigate their relationship. Two highly correlated (*r* = .534) factors with Eigenvalues higher than 1 emerged. The 9 DRSD items loaded on factor 1, while

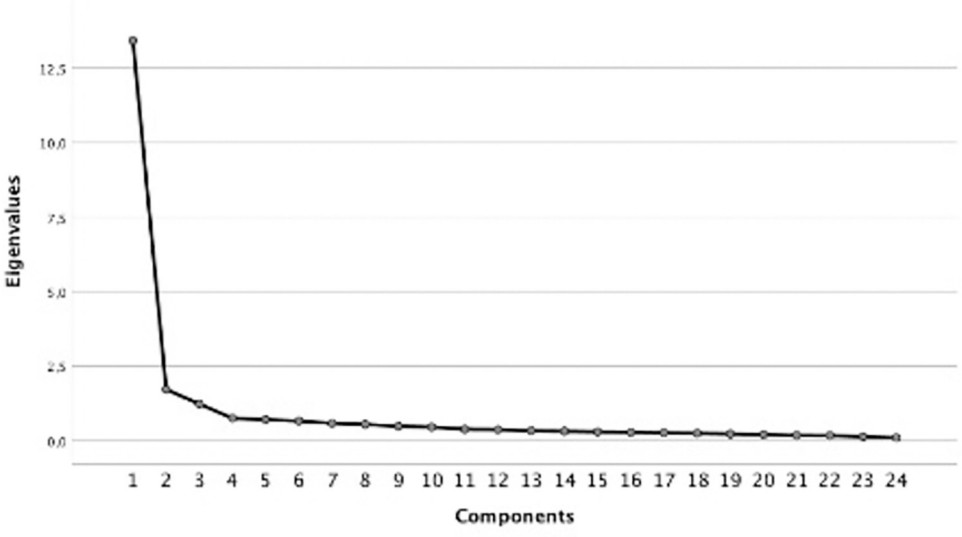

**Fig 2. Scree-plot of exploratory factor analysis.**

**Table 2. Correlations Study 1.**

| Variables | 1 | 2 | 3 | 4 | 5 | 6 | 7 | 8 |
|---|---|---|---|---|---|---|---|---|
| 1. DRSD | - | | | | | | | |
| 2. SDO | .473** | - | | | | | | |
| 3. RWA | .217** | .461** | - | | | | | |
| 4. Left/Right | .459** | .543** | .536** | - | | | | |
| 5. Liberal/Conservative | .378** | .397** | .420** | .605** | - | | | |
| 6. Social Darwinism | .606** | .681** | .552** | .580** | .395** | - | | |
| 7. Neoliberalism | .436** | .599** | .545** | .624** | .394** | .695** | - | |
| 8. Protective Behavior | -.629** | -.269** | .058 | -.193** | -.163** | -.361** | -.199** | - |

$N = 300$

** $p < .001$.

the items of the social Darwinism scale loaded on factor 2. These results support our conceptualization of DRSD as a structurally distinct part of general social Darwinism.

**Correlations of disease-related social Darwinism.** To test Hypotheses 1 a)–c), correlations between relevant variables were calculated (see, Table 2). Participants who scored higher on the DRSD scale also scored higher on the social Darwinism scale ($r = .606$, $p < .001$). Furthermore, participants with higher DRSD scores exhibited higher values of SDO ($r = .473$, $p < .001$), RWA ($r = .217$, $p < .001$), and neoliberal thinking ($r = .436$, $p < .001$).

To provide further evidence in support of our validity analyses, we performed a Harman's single factor test to investigate potential common method bias. The resulting single factor explained about 32% of variance, suggesting that common method bias is not a concern.

**Predicting disease-related social Darwinism.** To test Hypothesis 2, a linear regression with the left-right self-assessment as predictor and DRSD as the dependent variable was conducted. Participants' identification on the left-right axis significantly predicted DRSD ($\beta = .459$ $t(298) = 8.919$, $p < .001$), indicating that with increasing self-identification as politically right-wing, endorsement of DRSD also increased. The left-right axis explained about 21.1% of variance of the DRSD scale. A linear regression with party preference as the predictor yielded similar results (people who chose "other" or "none" were excluded from this analysis, $N = 236$); participants who preferred more right-wing parties tended to show higher scores of DRSD (see Table 3). We decided to calculate two separate regression analyses for left-right self-identification and party preference because even though the dimensions are correlated, someone identifying as more right-wing could prefer libertarian (economically right-wing), traditionalist/conservative (moderate right-wing) or populist right-wing parties. Hence, calculating two separate regression analyses should provide a more detailed picture.

**Predicting behavior during the COVID-19 pandemic.** To test Hypothesis 3 a), a hierarchical regression with self-reported behavior during the pandemic as the dependent variable was conducted. In Step 1, participants' score on the social Darwinism scale was added as the

**Table 3. Regression analyses predicting DRSD.**

| Dependent Variable | Predictors | β | B | 95% CI | $R^2$ | $R^2_{adj}$ |
|---|---|---|---|---|---|---|
| DRSD | Left/Right | .459 | .580 | .452, .708 | .211 | .208 |
| | Party Pref. | .466 | .256 | .194, .319 | .217 | .214 |

$N = 300$, $N_{PartyPref.} = 236$.

**Table 4. Regression analyses predicting protective behavior.**

| Dependent Variable | Predictors | β | B | 95% CI | $R^2$ | $R^2_{adj}$ | Tolerance |
|---|---|---|---|---|---|---|---|
| Protective Behavior | SD | -.361 | -.398 | -.515, -.281 | .131 | .128 | |
| | SD | .032 | .035 | -.088, .157 | | | .633 |
| | DRSD | -.648 | -.564 | -.662, -.467 | .397 | .393 | .633 |
| Protective Behavior | Left/Right | .291 | .321 | .111, .530 | | | .210 |
| | DRSD | -.416 | -.363 | -.603, -.122 | .416 | .410 | .100 |
| | Interaction | -.389 | -.065 | -.127, -.003 | | | .055 |
| | Party Pref. | .352 | .169 | .063, .275 | | | .208 |
| | DRSD | -.222 | -.193 | -.421, .034 | .392 | .385 | .150 |
| | Interaction | -.637 | -.054 | -.086, -.022 | | | .072 |
| | Left/Right © | .115 | .127 | .018, .235 | | | .786 |
| | DRSD © | -.659 | -.574 | -.662, -.485 | .416 | .410 | .741 |
| | Interaction | -.094 | -.065 | -.127, -.003 | | | .935 |
| | Party Pref. © | .015 | .007 | -.047, .062 | | | .777 |
| | DRSD © | -.638 | -.557 | -.657, -.457 | .392 | .385 | .774 |
| | Interaction | -.173 | -.054 | -.086, -.022 | | | .966 |

$N = 300$, $N_{PartyPref.} = 236$, mean-centered predictors are marked with ©, for these regression analyses the interaction is the product of the mean-centered predictors.

only predictor. In Step 2, DRSD was added as the second predictor. Model 1 was significant ($F$ (1, 298) = 44.787, $p < .001$); people with higher scores on the social Darwinism scale showed less precautionary behavior during the pandemic. Social Darwinism explained approximately 13.1% of variance in behavior. Model 2 was also significant ($F$ (2, 297) = 97.613, $p < .001$) and explained about 39.7% of variance, which represents a significant increase of roughly 26.6% explained variance (change in $F$ (1, 297) = 130.915, $p < .001$) through the addition of DRSD as a predictor (see Table 4 for all analyses predicting protective behavior).

To test Hypothesis 3 b), we computed an interaction variable as the product of party preference and DRSD, which was then used as one of the predictors in a multiple linear regression. The other two were DRSD and party preference (participants who chose the options "other" or "none" were excluded from this analysis). The dependent variable was self-reported behavior during the pandemic. The regression model was significant ($F$ (3, 232) = 49.952, $p < .001$), explaining 39.2% of variance. The interaction variable was a significant predictor ($β = -.637$; $t$ (232) = -3.328, $p = .001$). The negative values are obviously due to the coding of behavior. Introducing the interaction variable in a second step results in a significant increase in explained variance of 2.9% (*Change in F*(1, 232) = 11.078, $p = .001$). We centered the predictor variables and calculated the same regression analyses again to make the main effects interpretable. The interaction effect was obviously still significant ($β = -.173$; $t$ (232) = -3.328, $p = .001$). DRSD was a significant negative predictor of protective behavior during the pandemic ($β = -.638$; $t$ (232) = -10.964, $p < .001$), while party-preference was not ($β = .015$; $t$ (232) = .266, $p = .790$). Conducting the same analysis with the person's left-right identification instead of party-preference ($N = 300$) yields very similar results; the interaction is also significant ($p = .042$). This indicates that for people preferring more right-wing parties (and identifying as further right) the negative effect of DRSD on protective behavior during the pandemic is more pronounced.

## Discussion

Study 1 revealed encouraging results regarding the DRSD scale. The short DRSD scale showed great reliability ($\alpha = .95$) and validity as it related to similar political attitudes. As expected, DRSD increased with a higher preference for the political right. Importantly for the current

purpose, DRSD was found to be related but not identical to general social Darwinism. Further-more, the substantial correlations with social Darwinism and SDO, as well as the high inter-item correlations (s. online supplemental material) provide further justification for our con-structed items. Moreover, the findings support the notion that DRSD is one of the attitudes formed or changed during the pandemic that not only explains people's health-related behav-ior but also relates to a broader societal/political dimension. To corroborate these findings, we conducted a second, high-powered and preregistered study in a different cultural context using a representative sample.

## Study 2

As pointed out in the introduction, we conceptualized social Darwinism and hence DRSD in the context of social and political attitudes. Therefore, to establish validity, we think it is very impor-tant to test our scale in a broader political context and in association with broader political atti-tudes. Furthermore, we think that in the "pandemic age" disease-related attitudes are gaining importance and thus also relate to broader political attitudes. Perceived threat due to COVID-19, for example, has been associated with more anti-Asian prejudice [45] and moderated the effects of RWA on nationalism and anti-immigrant attitudes during the pandemic [46]. Hence, Study 2 was designed to locate DRSD in the broader political context and investigate its associations with broader political attitudes. It was part of a larger investigation using a representative US sample. Our main goal was to replicate and extend our findings in a preregistered large study.

Based on Study 1, we expected DRSD to be correlated with RWA, SDO (Hypothesis 1a), and the reactivated F-scale (Hypothesis 1b), which was included as another measurement of authoritarianism. Moreover, we expected DRSD to correlate with participants' sympathy for republican politicians and their self-identification as more right-wing and conservative (Hypothesis 1c). Most importantly, we hypothesized that DRSD would predict political atti-tudes classified as more conservative, even when SDO (a related concept, which has been shown to predict political conservatism; [47]) is controlled for (Hypothesis 2).

A relatively recent attempt to explain why people support right-wing populist politicians and parties has focused on the idea of relative need deprivation [22]. The theory posits that globalization and modernization have led to feelings of relative deprivation of certain psycho-logical needs (e.g., for certainty or a coherent identity). Right-wing populists seem to signal opportunities to cope with those deprived needs. We exploratively assessed DRSD in relation to relative need deprivation [21], with the expectation that frustrated needs would predict higher sympathy for republican candidates (Hypothesis 3a). Moreover, we hypothesized that this relationship would be moderated by DRSD since social Darwinism is often associated with right-wing attitudes and ideologies (see Study 1, Hypothesis 3b). Additionally, policies like the No Child Left Behind Act implemented by Republicans have been described as social Darwinist [7,9], the same goes for Donald Trump's worldview [48], so people with frustrated psychological needs should be even more inclined to sympathize with Republican politicians (like Trump) when they simultaneously show high scores of DRSD. We expected this interac-tion to be stronger in people with stronger specific relative deprivation (Hypothesis 3c).

### Method and materials

**Participants.**   The study was conducted between 28.01.2022 and 30.01.2020, using Prolific. The sample was representative of the US population in terms of age, gender, and ethnicity. In total 601 participants were recruited. Participants were asked if they were familiar with 10 poli-ticians presented to them (8 real, 2 fake). As described in the pre-registration, participants were excluded if they indicated to be familiar with one or both fake politicians. This would

indicate either a lack of knowledge about US politics or a lack of concentration. Both would have negative effects on the validity of our dependent variable. On this basis, 68 participants were excluded; hence, the final sample consisted of $N = 533$ ($M_{age} = 45.75$, SD = 16.2). $N = 266$ identified as female, while $N = 7$ either indicated they identified as another gender or did not indicate gender at all. 72% of the sample identified as White/Caucasian, 12.6% as African-American, 6.6% as Asian and 4.3% as Hispanic. A post-hoc sensitivity analysis revealed that our final sample of 533 participants had an 80% power to detect a minimum change in $R^2 <$ .001 (when adding one predictor to one or two other predictors). Again, as in Study 1, this sample can be considered sufficiently big in terms of stability of correlation coefficients [37].

**Disease-related social Darwinism.** If not indicated otherwise participants gave their response on a 6-point Likert scale ranging from 1 ("strongly disagree") to 6 ("strongly agree").

DRSD was assessed using a translated version of the 14-item scale from Study 1. The scale was translated by the authors and checked by native speakers (s. online supplemental materials). Items were presented in a fixed-randomized order. Responses were averaged so that higher scores indicated a stronger endorsement of DRSD. For the same reasons as in Study 1, we used the short version of the scale (results using the full scale are reported in the online supplemental materials). A confirmatory factor analysis indicated that one item was not useful in measuring DRSD (We think the translation was sub-optimal), so it was excluded from further analyses. Items were presented in a fixed-randomized order (For short 8-item scale: $\alpha$ = .96; $M$ = 2.36; $SD$ = 1.37).

**Right-Wing Authoritarianism.** RWA was assessed using the revised, short version of the RWA scale proposed by Zakrisson [49]. Items were presented in a fixed-randomized order. Responses were reverse-coded when needed and averaged so that higher scores indicated a stronger endorsement of authorities ($\alpha$ = .89; $M$ = 2.7; $SD$ = .95).

**Authoritarianism.** To investigate all facets of authoritarianism, we also used the reactivated F-scale [50]. This scale was designed to measure the remaining 6 subscales of the original F-scale [51] not captured with RWA. The scale consists of 20 items, which were presented in a fixed-randomized order. Items were averaged so that higher scores indicated higher approval of authoritarianism ($\alpha$ = .91; $M$ = 2.97; $SD$ = .89).

**Social Dominance Orientation.** Preference for social hierarchies was measured using the same scale as in study 1 [41]. Items were presented in a fixed-randomized order. Responses to the 8 items were reverse coded when needed and then averaged so that higher scores indicated a stronger preference for inequality between groups in society ($\alpha$ = .91; $M$ = 2.09; $SD$ = 1.09).

**Relative deprivation.** Need deprivation was measured using 36 items consisting of two subscales: general and specific need deprivation with 18 items each. Both subscales are comprised of three scales representing the three types of psychological needs that can be deprived (existential needs, epistemic needs, and identity needs, compare [52]) each measured using 6 items. Items were presented in a fixed-randomized order. Items were reverse-coded as needed and then averaged so that higher scores reflected more strongly frustrated needs ($\alpha_{general}$ = .83, $M_{general}$ = 3.71, $SD_{general}$ = .7; $\alpha_{specific}$ = .94, $M_{specific}$ = 3.06, $SD_{specific}$ = 1.12). The distinction between specific and general need deprivation was not pre-registered, because their relation was unknown at that time.

**Political orientation.** Several different measures were used to assess political orientation and behavior. Participants were asked to indicate their position on a scale ranging from left to right ($M$ = 2.93, $SD$ = 1.41) and on a scale ranging from liberal to conservative ($M$ = 2.94, $SD$ = 1.44). Preference for political parties was assessed separately for the Democratic Party and the Republican Party ($M_{Dem}$ = 3.48, $SD_{Dem}$ = 1.55; $M_{Rep}$ = 2.39, $SD_{Rep}$ = 1.5), and participants were asked who they voted for in the 2020 presidential election ($N_{Biden}$ = 298, $N_{Trump}$ = 99, $N_{non-voter}$ = 108).

Furthermore, participants were asked to indicate if they were familiar with 10 politicians (4 Democrats, 4 Republicans and 2 fake politicians), and then to indicate how strongly they agreed with the views of these politicians (politicians were not presented as options if participants indicated they did not know them). A preference score was calculated as *Preference = (Omar + Ocasio-Cortez)–(Trump + Taylor Greene) + 10*, resulting in a scale ranging from 0 (strong preference for Republicans) to 20 (strong preference for Democrats; $M$ = 14.07, $SD$ = 5.84). We have decided to calculate the preference for politicians score in this way because we think that it helps us cover the whole political spectrum and thus provide a more nuanced picture. By using more "extreme" politicians for each party, we cannot only differentiate between the moderate center but also between people more on the "extreme" ends of the political spectrum. A scale consisting of 17 political statements was used to measure people's political views. Items were reverse-coded when needed and averaged so that higher scores reflected more conservative political views ($\alpha$ = .94; $M$ = 2.32; $SD$ = 1.18). All items were presented in a fixed-randomized order.

**Demographic variables.**   In addition to the demographic variables automatically provided by Prolific, we assessed people's age, gender, ethnicity and their highest level of completed education.

**Other instruments.**   There were other measures used in this study that are mentioned here but are not relevant to the hypotheses but that may, however, be used for exploratory analyses. A scale consisting of 12 items measuring left-wing attitudes was assessed ($\alpha$ = .86; $M$ = 3.56; $SD$ = .94) and participants were told that they could choose a nonprofit organization to which $100 would then be donated. They had three options: NAACP (National Association for the Advancement of Colored People; $N$ = 172), American Red Cross ($N$ = 328), NRA (National Rifle Association; $N$ = 33). All items were presented in a fixed-randomized order.

**Procedure.**   Participants were welcomed and thanked for their participation before they received some general information about the study (topic, duration, data protection, etc.). On the same page, participants were also informed that after completing the survey they would be given the chance to donate $100 to one of three organizations. Following this, participants had to provide their Prolific ID, after which the reactivated F-scale, the left-wing attitudes scale, RWA and SDO were presented with items in a fixed-randomized order. On the following page, the frustrated needs scales were presented followed by items concerning political views. Participants were asked if they were familiar with 10 politicians (Biden, Ocasio-Cortez, Omar, Sanders, Trump, Taylor Green, Romney, Liz Cheney & two fake politicians). The following items (how strongly do you agree with the views of the following people) were only presented for those politicians that participants indicated they were familiar with. Party preference and voting behavior were assessed on the next page. The political attitude scale and the DRSD scale followed on separate pages. Afterward, participants had a choice between three organizations to which $100 would be donated to and finally, demographics were administered on the final page of the survey.

## Results

**Confirmatory Factor Analysis (CFA).**   We conducted confirmatory Factor Analysis using the package "lavaan" [53] in R [54] for the 14-item scale and the 9-item short scale to test for a one-factor solution (results for the 14-item scale can be found in the online supplemental materials). Cutoff values for Criteria of fit were based on Hu and Bentler [55] who proposed values close to .95 for CLI and TLI. Following Kenny et al. [56], we think that in our case RMSEA values should not be considered too important, since RMSEA tends to penalize models with lower df with a positive bias. Considering that our model only had one factor with few variables, we think a focus on CFI and TLI is justified. Nevertheless, we report RMSEA values

**Table 5. Correlations Study 2.**

| Variables | 1 | 2 | 3 | 4 | 5 | 6 | 7 |
|---|---|---|---|---|---|---|---|
| 1. DRSD | - | | | | | | |
| 2. SDO | .598** | - | | | | | |
| 3. RWA | .524** | .551** | - | | | | |
| 4. Left/Right | .591** | .609** | .667** | - | | | |
| 5. Liberal/Conservative | .587** | .606** | .682** | .943** | - | | |
| 6. F-Scale | .676** | .651** | .802** | .668** | .675** | - | |
| 7. Politician Preference | -.760** | -.676** | -.648** | -.834** | -.827** | -.708** | - |

$N = 533$

$N_{PolPref} = 322$

** $p < .001$, Politician Preference was coded such that lower scores indicate a stronger preference for Republican politicians.

for transparency. The 9-item short scale showed a relatively good fit to a one-factor solution (CFI = .962, TLI = .949, RMSEA = .112), however, one item did not seem to fit the factor (this was the case for both the 14-item and short scale), so it was excluded from further analyses. The 8 remaining items showed good fit to a one-factor solution (CFI = .974, TLI = .963, RMSEA = .105). The SRMR for both models looked proper according to Hu and Bentler [55], with SRMR = .031 for the 9-item model and even better for the 8-item model with SRMR = .021. Overall, the CFA supported the notion of the scale as a one-dimensional measurement.

**Correlates of DRSD.** To test Hypotheses 1 a)-c), correlations between the relevant variables were calculated (see Table 5). Correlations indicated again that participants scoring higher on the DRSD scale also showed higher scores of RWA ($r = .524$, $p < .001$) and SDO ($r = .598$, $p < .001$). The same can be said for the reactivated f-scale ($r = .676$, $p < .001$). Participants exhibiting higher scores of DRSD also showed a stronger preference for Republican politicians ($r = -.760$, $p < .001$; the negative correlation is obviously due to the calculation of the preference score), self-identification as more right ($r = .591$, $p < .001$) and more conservative ($r = .587$, $p < .001$). Hypotheses 1 a)-c) were supported.

Again, to provide further evidence in support of our validity analyses, we performed a Harman's single factor test to investigate potential common method bias. The resulting single factor explained about 32% of variance, suggesting that common method bias is not a concern.

**Predicting conservative political orientation.** To test Hypothesis 2, we conducted a linear regression with DRSD and SDO as the predictors and the 17-item political attitudes scale as the dependent variable. The model was significant, explaining 62.4% of variance ($F (2, 530) = 439.542$, $p < .001$). SDO ($\beta = .467$; $t (530) = 14.039$, $p < .001$) and DRSD ($\beta = .417$; $t (530) = 12.533$, $p < .001$) were both significant predictors of conservative political orientation. When this is calculated as a hierarchical regression with DRSD being added in step 2, the addition of DRSD as a predictor explains a significant 11.1% of variance in addition to SDO (change in $F (1, 530) = 157.088$, $p < .001$). We, therefore, found confirming evidence for Hypothesis 2 (for all regression analyses see Table 6).

**Table 6. Regression Analyses predicting conservative attitudes.**

| Dependent Variable | Predictors | β | B | 95% CI | R² | R²_adj | Tolerance |
|---|---|---|---|---|---|---|---|
| Conservative Attitudes | SDO | .716 | .776 | .712, .841 | .512 | .511 | |
| | SDO | .467 | .506 | .435, .577 | | | .642 |
| | DRSD | .417 | .359 | .303, .416 | .624 | .622 | .642 |

$N = 533$.

**Predicting preference for politicians.** To test Hypothesis 3 a) we calculated two linear regressions. The predictor was general or specific frustrated needs. General frustrated needs significantly predicted politician preference ($\beta$ = .176; $t$ (320) = 3.201, $p$ = .002), as did Specific frustrated needs ($\beta$ = -.799; $t$ (320) = -23.767, $p$ < .001). Hypothesis 3 a) was partly supported. To test Hypotheses 3 b) and c) we calculated interaction variables of DRSD and specific/general need deprivation as the product of the respective two predictors. We performed linear regressions with three predictors: DRSD, general (Model 1) or specific (Model 2) need deprivation, and the corresponding interaction variable. DRSD was a significant predictor in both models (Model 1: $\beta$ = -1.076; $t$ (318) = -5.750, $p$ < .001; Model 2: $\beta$ = -.212; $t$ (318) = -1.978, $p$ = .049), general need deprivation were not ($\beta$ = .032; $t$ (318) = .483, $p$ = .629), specific needs deprivation were ($\beta$ = -.420; $t$ (318) = -5.464, $p$ < .001). The interaction variable was not significant in both models (Model 1: $\beta$ = .339; $t$ (318) = 1.766, $p$ = .078; Model 2: $\beta$ = -.252; $t$ (318) = -1.659, $p$ = .098; when calculated with the 14-item scale the interaction of DRSD and general frustrated needs was significant (s. online supplemental materials)). When added in an extra step the interaction variable explained 0.4% of additional variance in Model 1 and 0.3% of additional variance in Model 2. Hypotheses 3 b) and c) were not supported.

Because some of the betas are higher than 1 which is most likely due to collinearity being high, we decided to recalculate the regression analysis with mean-centered variables. Therefore, we mean-centered DRSD, general frustrated needs and specific frustrated needs before calculating their interaction variables as described above. These analyses were not pre-registered since we did not expect such high collinearity. Again, DRSD was a significant predictor in both models (Model 1: $\beta$ = —.754; $t$ (318) = -21.25, $p$ < .001; Model 2: $\beta$ = -.353; $t$ (318) = -7.514, $p$ < .001). General need deprivation ($\beta$ = .140; $t$ (318) = 3.884, $p$ < .001), and specific needs deprivation were both significant ($\beta$ = -.510; $t$ (318) = -11.065, $p$ < .001). The interaction variable was not significant in both models (Model 1: $\beta$ = .064; $t$ (318) = 1.766, $p$ = .078; Model 2: $\beta$ = -.060; $t$ (318) = -1.659, $p$ = .098). See Table 7 for detailed results.

**Explorative analyses.** The following analyses are not mentioned in the pre-registration as they comprise exploratory investigations of our data and should be interpreted as such. The measurements we use, however, are mentioned in the pre-registration and the Measures section of Study 2.

**Table 7. Regression Analyses predicting politician preference.**

| Dependent Variable | Predictors | $\beta$ | $B$ | 95% CI | $R^2$ | $R^2_{adj}$ | Tolerance |
|---|---|---|---|---|---|---|---|
| Politician Preference | Frustrated Needs (gen) | .176 | 1.454 | .560, 2.347 | .031 | .028 | |
| | Frustrated Needs (spec) | -.799 | -4.240 | -4.591, -3.889 | .638 | .637 | |
| | Frustrated Needs (gen)<br>DRSD<br>Interaction | .032<br>-1.076<br>.339 | .262<br>-4.709<br>.380 | -.806, 1.330<br>-6.320, -3.097<br>-.043, .804 | .598 | .594 | .292<br>.036<br>.034 |
| | Frustrated Needs (spec)<br>DRSD<br>Interaction | -.420<br>-.212<br>-.252 | -2.230<br>-.927<br>-.202 | -3.033, -1.427<br>-1.848, -.005<br>-.441, .037 | .704 | .701 | .157<br>.081<br>.040 |
| | Frustrated needs (gen) (C)<br>DRSD (C)<br>Interaction | .140<br>-.754<br>.064 | 1.159<br>-3.298<br>.380 | .572, 1.747<br>-3.605, -2.991<br>-.043, .804 | .598 | .594 | .967<br>.995<br>.968 |
| | Frustrated needs (spec) (C)<br>DRSD (C)<br>Interaction | -.510<br>-.353<br>-.060 | -2.705<br>-1.543<br>-.202 | -3.186, -2.224<br>-1.947, -1.139<br>-.441, .037 | .704 | .701 | .439<br>.423<br>.722 |

$N_{PolPref}$ = 322; Politician Preference was coded such that lower scores indicate a stronger preference for Republican politicians, mean-centered predictors are marked with ©, for these regression analyses the interaction is the product of the mean-centered predictors.

To investigate the role of DRSD in participants' self-reported voting behavior in the 2020 presidential election we performed a binominal logistical regression with DRSD as the predictor. The dependent variable was recoded so that voting for Donald Trump (1; $N$ = 99) was compared to voting for Joe Biden (0, $N$ = 298), while other answers to the original item were coded as missing. The regression model was significant ($\chi^2$ (1) = 133.463, $p <$ .001). The model explained between 28.6% (Cox & Snell $R^2$) and 42.3% (Nagelkerkes $R^2$) of variance. The model had an overall percentage of accurate classification of 82.9%, with a sensitivity of 51.5% and specificity of 93.3%. DRSD significantly predicted voting for Donald Trump ($p <$ .001), leading to a higher probability of voting for Donald Trump (vs. Joe Biden; OR = 3.031). We calculated the same binominal logistical regression and controlled for RWA and SDO as additional predictors. All correlations of the predictors were lower than $r$ = .8, indicating that multicollinearity was not a confounding factor in this model [57]. We added RWA and SDO in the first step and DRSD in the second step. Adding DRSD to the model increased the overall percentage of accurate classification by 2.5% and increased sensitivity by 7.1%. DRSD leads to a higher probability of voting for Donald Trump (vs. Joe Biden; OR = 1.943) even when controlling for RWA and SDO.

We calculated similar binominal logistical regressions with participants' donation behavior as the dependent variable, coded as 0 (donating to the NAACP, $N$ = 172) and 1 (donating to the NRA, $N$ = 33). The model including just DRSD was significant ($\chi^2$ (1) = 112.638, $p <$ .001), explaining between 42.3% (Cox & Snell R2) and 72.1% (Nagelkerkes R2) of variance. The model had an overall percentage of accurate classification of 91.7%, with a sensitivity of 72.7% and specificity of 95.3%. DRSD significantly predicted donating to the NRA ($p <$ .001), leading to a higher probability of donating to the NRA (vs. the NAACP; OR = 6.944). Adding DRSD to RWA and SDO in a second step increased the overall percentage of accurate classification by 3.9% and increased sensitivity by 9.1%. DRSD leads to a higher probability of donating to the NRA (vs. the NAACP; OR = 3.875) even when controlling for RWA and SDO.

We checked for possible outliers using both leverage (values higher than .2 [58]) and Cooks-distance (values higher than 1 [59]) in all four models. Only in the model with DRSD, SDO, and RWA predicting donation behavior we found some outliers based on leverage but not based on Cooks-distance, which is why we decided to not exclude them.

## Discussion

Investigating DRSD in a large representative US sample confirmed the findings of the previous study. Across both investigations, DRSD scale was found to be a valid measurement of disease-related social Darwinist thinking, a specific form of social Darwinist thinking particularly relevant in the pandemic. Furthermore, Study 2 indicated that the scale was useful in a different cultural and political context, as an important construct in explaining more generalized political attitudes outside of COVID-19 related political behaviors. Exploratory analyses point to a possible role of DRSD in voting decisions and deciding to support certain organizations like the NRA.

## General discussion

The COVID-19 pandemic has changed the way people think and live together in societies. If experts are correct in predicting that ecological issues like deforestation and extinctions make future pandemics ever more likely [60], we might already live in a pandemic world. This highlights the importance of understanding how political attitudes are formed and changed during pandemics and how they relate to societal life.

The present paper aims at addressing this topic by introducing Disease-Related Social Darwinism as a disease-related belief characterized by the wish to let nature run its course for

progress to occur. We generally defined social Darwinism as an application of Darwin's theory of natural selection to the development of human societies. Specifically, social Darwinism is characterized by the idea that inequality is inevitable and preferable because it is a result of natural selection. According to this logic, natural selection equals societal progress because, if permitted, the weaker members of a group are weeded out and the group is strengthened accordingly. Hence, diseases (particularly pandemics) are perceived as a natural engine of evolution and progress since, if not controlled or counteracted, they would kill mostly the supposedly "weaker" members of a group. Therefore, DRSD, as a specification of general social Darwinism, provides justification for politicians not to protect vulnerable members of society and for citizens to not protect themselves and others. To address this way of thinking, we investigated the quality and usefulness of the DRSD scale. Results of two studies indicate that the scale is reliable, valid and useful in predicting people's protective behavior and their political attitudes.

In Study 1 (conducted in Germany), we investigated DRSD in the context of the COVID-19 pandemic. Besides great reliability, the DRSD scale showed significant relations with established political measures such as general social Darwinism, RWA, SDO and neoliberalism thus indicating the scale's validity. The high inter-item correlations (s. online supplemental material) in both studies also justify the constructed items. Most importantly, DRSD significantly predicted people's behavior during the pandemic, even when general social Darwinism [5] was controlled for. People endorsing a view of diseases as a natural engine of evolution reported less protective behavior during the pandemic. DRSD also moderated the relation between people's party preference and behavior. People sympathizing with more right-wing parties showed less protective behavior, especially if they also had higher scores of DRSD.

Preregistered Study 2 confirmed the reliability and validity of the DRSD scale in a different national (i.e., US) and representative sample. Correlations with SDO and RWA were replicated, and other significant correlations corroborated the validity of the scale. The central finding is that DRSD significantly predicted people's political attitudes, after controlling for SDO, which has been linked to more conservative attitudes [27]. People who perceive diseases in a social Darwinist way tended to exhibit more conservative political attitudes.

With our findings, we build on the fundamental work by Saud [26] and Rudman and Saud [5] investigating social Darwinist beliefs in political contexts. We replicate central findings from their studies concerning the relationships between social Darwinism, SDO and RWA. However, most importantly, contributing to the social Darwinism framework, we introduce a new measurement of social Darwinist beliefs about diseases, which is not only reliable and valid but also very economical (8-items for the English short scale).

## Applications, limitations and avenues for future research

The studies presented here provide a solid basis for future research on the role of DRSD and its association with political attitudes and behavior. Considering that social Darwinist ideas arose in western societies during the Gilded Age (Degler, as cited in [5]), future research should investigate social Darwinist beliefs in non-western cultures to better understand their cultural aspect.

We also believe that the DRSD scale contributes to the landscape of political attitude measures because it provides answers to basic societal questions such as how societal progress should occur and whether equality is preferable. More generally, it touches on the foundational tenets of how individuals perceive human life and the question of whether human beings should be considered as rather similar or quite different from each other. Furthermore, until now there has not been a measurement directly capturing the social Darwinist idea that

diseases can serve a positive function for society. The incremental validity our scale showed compared to social Darwinism and SDO in predicting political behavior and orientation proves that our scale captures a specific form of social Darwinist attitudes not adequately measured by existing measures. Exploratory analyses in Study 2 point to a possible role of DRSD in voting decisions and more directly behavioral decisions relating to donating $ 100 to a specific organization. DRSD increased the likelihood of reporting having voted for Donald Trump (vs. Joe Biden) in the 2020 election and wanting to donate to the NRA (vs. NAACP). These analyses provide fertile ground for future research on DRSD as a social/political attitude. For exploratory reasons we also investigated the relationship between DRSD and general/specific relative deprivation which resulted in an inconsistent pattern. Hence, the motivational underpinnings of DRSD need further investigation.

Regarding the construction of the scale, we found that all reverse-coded items loaded on a different factor. This might be due to the fact it is difficult to define opposites to social Darwinist beliefs. Nevertheless, this aspect should be addressed in future studies. Additionally, the CFA in Study 2 yielded high values of RMSEA. Even though we think that in our case this should not be considered a great threat to model fit [56], especially considering the proper CFI, TLI, and SRMR values, future research should nonetheless investigate the structure of our scale further.

**Is disease-related social Darwinism an individual, intergroup, or societal phenomenon?.**   Our results suggest that DRSD justifies ideas of inequality on almost every social dimension, starting from the very individual and reaching to societal levels. On the individual dimension, DRSD relates to people's behavior and their conceptions of what is right and wrong (e.g., "Should I protect vulnerable members of my group?"). On the intragroup dimension, DRSD justifies a hierarchical (vs. egalitarian) organization of an individual's own ingroup, as indicated by the significant correlations with RWA in both studies. As already shown by Rudman and Saud [5] and replicated in the present study, social Darwinism and hence DRSD justify why certain groups should dominate others in society, as illustrated by the significant correlations with SDO. From the significant prediction of conservative political attitudes, it can be assumed that DRSD also plays a role on a societal dimension, justifying an individual's perception of policy decisions and party/politician preference. We think that the multi-dimensional influence DRSD seems to have is a good argument to consider it as a foundational attitude underlying conservatism and right-wing political attitudes.

**How useful is the DRSD scale outside of the pandemic?.**   In the present investigation, DRSD significantly predicted people's self-reported protective behavior during the pandemic. Showing that people, who perceive diseases as having a positive function for societies, reported less self-protective behavior during the COVID-19 pandemic. A recent study confirmed that self-reported measures of social distancing (even short two-item measures like the one we used) are valid measures of actual social distancing behavior during the pandemic, assessed with GPS tracking [61]. Nevertheless, it can be asked whether the DRSD would be useful outside of the context of COVID-19 and the accompanying pandemic. Even if experts predicting a pandemic future are not correct, the COVID-19 pandemic has demonstrated that people's disease-related attitudes are not only relevant to an individual's health but also have a great impact on a societal/political dimension, being used to popularize certain ideologies or positions. With the DRSD scale we provide a useful tool to capture such an attitude, finally used to justify the death of vulnerable people. Individuals holding social Darwinist beliefs perceive diseases as an opportunity for growth of the group/society rather than a risk. Hence, vaccinations and other treatments preventing severe illness may be rejected as threats to human progress even outside the current pandemic context. Accordingly, DRSD may have a great impact on a societal level, since it can be a reason for people's vaccine hesitancy, which can cause broader

societal problems [16,17]. Furthermore, DRSD can serve as a justification for political decisions regarding diseases which makes it an interesting attitude to consider in the realm of political orientation. Moreover, the already-mentioned incremental validity our scale possesses in comparison to existing measurements, demonstrates that our scale is relevant for future research on people's disease-related attitudes and their effects. Such future research should investigate the relation of DRSD to actual disease-related behavior by using, for example, smartphone location data or step-tracking devices to assess social-distancing behavior.

## Author Contributions

**Conceptualization:** Paul Nachtwey.

**Data curation:** Paul Nachtwey, Eva Walther.

**Formal analysis:** Paul Nachtwey, Eva Walther.

**Methodology:** Paul Nachtwey, Eva Walther.

**Supervision:** Paul Nachtwey, Eva Walther.

**Validation:** Paul Nachtwey, Eva Walther.

**Writing – original draft:** Paul Nachtwey, Eva Walther.

**Writing – review & editing:** Paul Nachtwey, Eva Walther.

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
