## [Decision Letter · Decision Letter 0]

22 Sep 2022

PONE-D-22-19515Survival of the fittest in the pandemic age: Introducing Disease-Related Social DarwinismPLOS ONE

Dear Dr. Nachtwey,

Thank you for submitting your manuscript to PLOS ONE. After careful consideration, we feel that it has merit but does not fully meet PLOS ONE’s publication criteria as it currently stands. Therefore, we invite you to submit a revised version of the manuscript that addresses the points raised during the review process. I was able to get one evolutionary psychologist and one prejudice researcher with EP leanings. Both identified major issues that they think warrant your attention prior to publication but they suggest there is "light at the end of the tunnel" for this paper.

We look forward to receiving your revised manuscript.

Kind regards,

Peter Karl Jonason

Academic Editor

PLOS ONE

Journal Requirements:

2. You indicated that ethical approval was not necessary for your study. Could you please provide further details on why your study is exempt from the need for approval and confirmation from your institutional review board or research ethics committee (e.g., in the form of a letter or email correspondence) that ethics review was not necessary for this study? Please include a copy of the correspondence as an ""Other"" file.

"The authors did not receive funding for this work."

Reviewers' comments:

Reviewer's Responses to Questions

**Comments to the Author**

1. Is the manuscript technically sound, and do the data support the conclusions?

Reviewer #1: Partly

Reviewer #2: Yes

2. Has the statistical analysis been performed appropriately and rigorously? 

Reviewer #1: No

Reviewer #2: Yes

3. Have the authors made all data underlying the findings in their manuscript fully available?

Reviewer #1: Yes

Reviewer #2: Yes

4. Is the manuscript presented in an intelligible fashion and written in standard English?

Reviewer #1: Yes

Reviewer #2: Yes

5. Review Comments to the Author

Reviewer #1: The manuscript entitled „Survival of the fittest in the pandemic age: Introducing Disease-Related Social Darwinism” describes two studies on the Disease-Related Social Darwinism (DRSD) as a new concept inspired by the changes in societal attitudes toward protective measures and politics during the COVID-19 pandemic. The Authors described this construct (DRSD) in the context of sociopolitical attitudes (SDO; RWA; SD) and examined the expectations regarding political preferences related to this belief (DRSD). Since this new construct seems to be interesting development of social Darwinism, I have some suggestions concerning the description of the studies and its methodology.

#1. The Authors tested a lot of hypotheses concerning various groups of variables possibly corelated with DRSD. I think that the introduction would benefit from dividing it into sections concerning the associations between groups of such variables and DRSD (e.g., Political preferences and DRSD; DRSD and the COVID-19 preventive measures). The Authors could also include a figure in order to illustrate their hypotheses regarding DRSD. I could help in making the manuscript easier to follow.

#2. The abstract includes the sentences which are the copies of sentences present within the manuscript. I suggest to correct it in order to avoid repetitions.

#3. Throughout the manuscript the Authors are using wording suggesting causal role of the DRSD for political attitudes (e.g., p. 2; l. 29; p. 16; l. 344-346). Their studies are cross-sectional and correlational. Such a design gives no argument for causal interpretations.

#4. The first paragraphs of the introduction should be concise and avoid unnecessary references. I think that the Authors could simply introduce the DRSD and illustrate the beliefs similar to DRSR during the pandemic with one example. I am also wondering whether personal references to politicians are necessary.

#5. The suggestions that the DRSD beliefs “have been formed or changed during the pandemic” could be premature. The statements that arose during the pandemic and seem to be the examples of the DRSD could be better see globally, but could be previously present e.g. in individual’s attitudes toward the health care systems in their countries. The sentences such as “although social Darwinism has been widespread in societies for a long time, it can be assumed that the COVID-19 pandemic has increased the prevalence of this thinking” better express the possible situation associated with the DRSD beliefs.

#6. Study 1 has numerous goals. Moreover, the Authors stated: “Most importantly, however, we want to show that the scale has incremental validity in comparison to the standard social Darwinism scale when predicting people’s protective behavior during the COVID-19 pandemic (Hypothesis 3a)”. In my opinion, the central goal of the study 1 was to develop the measure of the DRSD. Thus, inspection of its internal structure and validity should be the main goal of the Study 1.

#7. The better justifications of sample size in both studies are necessary. The sentences such as “We decided to recruit around 300 participants based on Rudman and Saud (2020), who recruited around 400 participants for most of their studies but also computed more complex calculations than we planned for” are not enough. How the number of the participant could be described in the context of the measure development? Or in the context of stability of correlation coefficients?

#8. Better descriptions of exclusion and inclusion criteria are missing in both studies.

#9. Did the Authors use also the parallel analysis in order to inspect the correct number of factors in the DRSD measure?

#10. According to the Authors hypothesis that the DRSD is a variant of SD, the hypotheses about associations between the DRSD, SDO, and RWA, should be tested by using partial correlation (controlled for SD). SDO and SD are correlated stronger that the DRSD and SDO. The Authors should also mention such results in the discussion. In my opinion some CFA testing whether the DRSD and SD are different construct could be useful.

#11. The justification of using separate RAs to examine the associations between party preferences and left/right political orientation should be provided in the manuscript. Which were the correlations between these variables? In general, the authors can provide statistics concerning collinearity in their RA analysis.

#12. The Authors used probably a hierarchical regression (see l. 312). Did they centered variables before calculating interactions? Were interactions entered into regression models in separate step? Did that step introduce a significant change in R square? How were the interactions unpacked?

#13. The description of the goals of Study 2 introduces new construct which were not previously mentioned in the introduction (needs deprivation). I suggest to elaborate on these predictions in separate section in the introduction. In both studies some moderations appeared as a goals of the examinations. However, they were not elaborated and introduce properly in the introduction. Such a situation create a sense of chaotic argumentation and multiple goals which the Authors try to obtained using the one sample.

#14. The political orientation measure used in Study 2 should be better described. It consists of 4 Democrats and 4 Republicans, but the calculation is based only on 2 of each category. Please, justify this score calculation. Please also include some data on the validity.

#15. The other instruments again create a feeling that the Authors include many measures in their study and tried to find some significant associations with the DRSD. It would be better to clearly justify why these instruments were used. I am wondering why the Authors did not controlled for SD in the study 2? Moreover, the study 2 is less associated with the exact wording of the title, which suggest testing the DRSD in the context of the pandemic. Study 2 is more about the associations between the DRSD and political preferences.

#16. The CFA brought inconsistent results. Both CFI and TLI seem proper (but please provide also criteria of fit which were used), but RMSEA is clearly problematic. Moreover, the Authors mention two-factor solution, which was not previously described. Did the Authors tested the differences between chi square of both one- and two-factor solutions?

#17. The hierarchical RA in Study 2 looks problematic. First, when the goals is to analyze interaction, the Authors have to state how they computed interaction term, and did he model explained more variance when the interaction term was entered. Some standardized betas are higher than 1.00 – please explain such situation. Again, collinearity statistics are necessary.

#18. The sections entitled “Is social Darwinism an individual, intergroup, or societal phenomenon?” seem misleading in the discussion. The studies were about the DRSD, not the SD itself. Thus, I suggest to keep to the results of the Authors’ studies. However, the structure of this section could be used to structure the introduction in terms of correlates of the DRSD.

Reviewer #2: I would like to thank for possibility to review this paper on subject of creation of disease-specific social Darwinism scale.

Even general social Darwinism is a rather narrow variable, disease-specific social Darwinism very much so. I think that constructing such a measurement tool, very specific in scope, brings limited novelty to the literature. In my opinion, such a niche scale should be of utmost quality in order to be consider worthy of publication.

Theoretical part of the article is well-written and concise; the same could be said about discussion. Scale has good reliability. Method chosen for testing scale’s validity are proper, but Authors do not test validity by any other way than other questionnaires, which is unfortunate. The same applies to predicting behavior during pandemic “over and above” general social Darwinism. It is great idea, but unfortunately behavior was measured by 2 simple, self-report questions.

In my opinion, authors should implement at least one non-self-report measure to check scale’s validity. It is harder to do, but if scales are validated only by other paper-and-pencil measures, we cannot be fully sure what is their relation to real world.

In summary, I think this article is just not giving enough: in case of small, narrow contribution, it should met the highest standards. I would recommend publishing this article only if authors provide non-paper-and-pencil proves of scale validity and predictive utility (which may require additional study).

I also have two minor comments:

1. I couldn’t find information about national identity of the participants of the first study; Authors inform in the abstract that they collected data from two nationalities, but it is not evident what is the first one

2. Beta’s should not be written in italics

6. PLOS authors have the option to publish the peer review history of their article (what does this mean?). If published, this will include your full peer review and any attached files.

Reviewer #1: **Yes: **Marcin Moroń

Reviewer #2: **Yes: **Jarosław Piotrowski

---

## [Author Response · Author response to Decision Letter 0]

8 Nov 2022

Dear Dr. Jonason and dear Reviewers,

we would like to thank you for the constructive and helpful feedback on our manuscript. We are very happy for the opportunity to resubmit a revised manuscript version.

We believe that the reviewers' and your comments have helped us substantially to improve our manuscript and we were particularly grateful for the many concrete suggestions. Based on these comments and suggestions, we have made numerous changes to the manuscript, which are all detailed in the responses (in italics) below. We think that we were able to address all of the raised concerns. However, if anything remains unsatisfactory, we will happily undertake more revisions.

We are looking forward to hearing from you! And thanks again.

Sincerely,

Paul Nachtwey & Eva Walther

Note: the line numbers refer to the unmarked manuscript version, as the line numbers are subject to change in the marked-up version of the document depending on how changes are made visible in Microsoft Word. 

Journal Requirements

1. You indicated that ethical approval was not necessary for your study. Could you please provide further details on why your study is exempt from the need for approval and confirmation from your institutional review board or research ethics committee (e.g., in the form of a letter or email correspondence) that ethics review was not necessary for this study? Please include a copy of the correspondence as an ""Other"" file.

We appreciate the concern of PLOS ONE for ethical research. Separate ethical approval from the University was not necessary for our study because we strictly followed the ethical guidelines of the German Society for Psychology (DGPs; Deutsche Gesellschaft für Psychologie, 2016) to conduct ethical psychological research. Hence, we received an “Exempt from Ethics” from the ethics committee of the Universität Trier which we uploaded as an extra file with the revised version of our manuscript. 

"The authors did not receive funding for this work."

The Authors received no specific funding for this work

Reviewer 1

1. The Authors tested a lot of hypotheses concerning various groups of variables possibly corelated with DRSD. I think that the introduction would benefit from dividing it into sections concerning the associations between groups of such variables and DRSD (e.g., Political preferences and DRSD; DRSD and the COVID-19 preventive measures). The Authors could also include a figure in order to illustrate their hypotheses regarding DRSD. I could help in making the manuscript easier to follow.

We thank Reviewer 1 for this helpful comment. We agree that the number of hypotheses with different variables made the introduction difficult to follow. We thus included more subheadings in the introduction section (see, l. 82, 105, 128, 153) and we have added a part about DRSD and political orientation for clarification (see, l. 105-127). 

Based on the suggestion from the reviewer, we have now added a figure (see, l. 194) illustrating the hypotheses of Study 1 because we agree that this will enhance the comprehensibility of the manuscript and theoretical assumptions respectively. This figure can also be found at the end of this Letter.

2. The abstract includes the sentences which are the copies of sentences present within the manuscript. I suggest to correct it in order to avoid repetitions.

We appreciate this suggestion from Reviewer 1. We have changed the sentences in our abstract to avoid repetition (see, l. 25-30, 34f). 

3. Throughout the manuscript the Authors are using wording suggesting causal role of the DRSD for political attitudes (e.g., p. 2; l. 29; p. 16; l. 344-346). Their studies are cross-sectional and correlational. Such a design gives no argument for causal interpretations.

We are thankful for these observations. For obvious reasons, we did not want to create the idea of causality in our manuscript, so we went through the whole manuscript and carefully changed every instance of misleading wording (see, l. 28, 95, 382, 650, 674, 692, 725).

4. The first paragraphs of the introduction should be concise and avoid unnecessary references. I think that the Authors could simply introduce the DRSD and illustrate the beliefs similar to DRSR during the pandemic with one example. I am also wondering whether personal references to politicians are necessary.

We agree with the reviewer’s critique that the first paragraph of our introduction may have been lengthy. We have thus shortened it considerably and included as few references as possible (see, l. 49-60). We also dropped references to individual politicians. As the reviewer has suggested, we have illustrated the beliefs similar to DRSD with one example (see, l. 56f). We have added parts of the deleted introduction to the part about DRSD and political Orientation (see, l. 109-114). 

5. The suggestions that the DRSD beliefs “have been formed or changed during the pandemic” could be premature. The statements that arose during the pandemic and seem to be the examples of the DRSD could be better see globally, but could be previously present e.g. in individual’s attitudes toward the health care systems in their countries. The sentences such as “although social Darwinism has been widespread in societies for a long time, it can be assumed that the COVID-19 pandemic has increased the prevalence of this thinking” better express the possible situation associated with the DRSD beliefs.

Thanks for this comment. We did not intend to raise premature conclusions. Due to the restructuring of our opening paragraph (see point 4 raised by Reviewer 1), the critical sentence 'have been formed or changed during the pandemic' is no longer part of our manuscript. However, because there is empirical evidence supporting the sentence: ‘although social Darwinism has been widespread in societies for a long time, it can be assumed that the COVID-19 pandemic has increased the prevalence of this thinking’ this part remains in the manuscript (see, l. 83f).

6. Study 1 has numerous goals. Moreover, the Authors stated: “Most importantly, however, we want to show that the scale has incremental validity in comparison to the standard social Darwinism scale when predicting people’s protective behavior during the COVID-19 pandemic (Hypothesis 3a)”. In my opinion, the central goal of the study 1 was to develop the measure of the DRSD. Thus, inspection of its internal structure and validity should be the main goal of the Study 1.

We agree with the Reviewer’s comment that the most important goal of Study 1 is the inspection of structure and validity of our scale. Hence, we reformulated the respective sentences accordingly (see, l. 169f, 189). 

7. The better justifications of sample size in both studies are necessary. The sentences such as “We decided to recruit around 300 participants based on Rudman and Saud (2020), who recruited around 400 participants for most of their studies but also computed more complex calculations than we planned for” are not enough. How the number of the participant could be described in the context of the measure development? Or in the context of stability of correlation coefficients?

We appreciate the Reviewer’s concern regarding the justification of the sample size. 

For Study 1, we decided to recruit as close to the sample size of Rudman and Saud (2020) as possible. A post hoc sensitivity analysis revealed that a sample of 300 participants has an 80% power of detecting a minimum Change in R2 = .001 (when investigating one predictor added to one or two other predictors). It has also been shown that correlation coefficients stabilize around 250 participants (Schönbrodt & Perugini, 2013), and that 200 participants are sufficient for factor analyses with less than 40 variables (our item-pool consisted of 24 items; Comrey, 1988). Hence, for the main goal of inspecting internal structure and validity of our scale (using EFA, correlational and regression analyses), 300 participants should be sufficient. 

Regarding Study 2, post-hoc sensitivity analyses revealed that a sample of 533 participants has an 80% power of detecting a minimum Change in R2 < .001 (when investigating one predictor added to one or two other predictors). 

To be more transparent, we have now added these explanations in part to the manuscript (see, l. 203-209, 433-437). 

8. Better descriptions of exclusion and inclusion criteria are missing in both studies.

Again, we sympathize with the concern of Reviewer 1 about transparency concerning the sample size. 

For Study 1, based on pre-testing of the study we decided that the minimum time it takes to at least read all items is around 5 minutes. Therefore, 4 participants that took less than 5 minutes were excluded from the sample. This more detailed description has also been added to the manuscript (see, l. 209-211). 

For Study 2, participants were presented with 10 politicians (8 real + 2 fake ones) and then asked to indicate if they knew these politicians. Participants were excluded if they expressed, they were familiar with one or both fake politicians because this implies either a lack of concentration or a serious lack of knowledge regarding US politics that might lead to flawed results. This more detailed justification has also been added to the manuscript (see, l. 425-430). 

9. Did the Authors use also the parallel analysis in order to inspect the correct number of factors in the DRSD measure?

We thank Reviewer 1 for the reminder about this possibility. However, after reinspecting the scree plot, we were positive that it very clearly suggests a one-factor solution and thus no parallel analysis would be necessary. To make this decision more transparent we have added the scree-plot as a figure to our manuscript (see, l. 314). We have also added it at the end of this letter for clarification. 

10. According to the Authors hypothesis that the DRSD is a variant of SD, the hypotheses about associations between the DRSD, SDO, and RWA, should be tested by using partial correlation (controlled for SD). SDO and SD are correlated stronger that the DRSD and SDO. The Authors should also mention such results in the discussion. In my opinion some CFA testing whether the DRSD and SD are different construct could be useful.

We thank Reviewer 1 for this comment. However, based on our theoretical reasoning, we assume that DRSD is a specification of the broader concept of Social Darwinism applied to the realm of diseases (see added Figure 1). Hence, partializing out Social Darwinism or computing the proposed CFA would not be appropriate. We have added a new paragraph into the introduction to make this consideration clearer and we also refer to this in the discussion section (see, l. 92f, 126f, 171ff, 660f). 

11. The justification of using separate RAs to examine the associations between party preferences and left/right political orientation should be provided in the manuscript. Which were the correlations between these variables? In general, the authors can provide statistics concerning collinearity in their RA analysis.

Thanks for this important comment. Because only one predictor was used in relevant RA, we think that collinearity concerns are unfounded. However, we added collinearity statistics for regression with multiple predictors (see Tables 4,6,7). 

We decided to calculate two separate RAs because we think that differentiating between self-placement on a left-right spectrum and preference for political parties in our RAs provides a more nuanced picture of the results. This is because, previous research indicates that these dimensions do not necessarily measure the same construct (Lesschaeve, 2017). Even though the dimensions are correlated (r = .396 when no participants are excluded, r = .706 when participants selecting no or another party were excluded), someone self-identifying as more right-wing could prefer libertarian parties (economically right-wing), traditionalist-conservative parties (moderate right-wing), or populist right-wing parties (van Assche et al., 2019). So even though there is overlap between these variables, they are by no means identical. This more detailed justification has been added to the manuscript (see, l. 334-338).

12. The Authors used probably a hierarchical regression (see l. 312). Did they centered variables before calculating interactions? Were interactions entered into regression models in separate step? Did that step introduce a significant change in R square? How were the interactions unpacked?

Thank you for raising these questions. The reviewer’s assumption is correct we used a hierarchical regression. We apologize for the mistake in wording and corrected the term throughout the manuscript (see, l. 343, 561). In the previous version of the manuscript, we did not center variables for the regression analyses since we were mainly interested in the interaction (centered variables do not change results for the interaction analyses). Based on your comment, we additionally calculated the regression analyses with centered variables to be able to interpret the main effects (instead of calculating two separate RAs as we did in the previous version of the manuscript). These additional analyses have been added to the manuscript (see, l. 365-369). 

Based on your suggestion, we have also added multicollinearity statistics for regression analyses with multiple predictors to our manuscript (see e.g., Table 5). 

We added interaction terms in an additional step; this is now also mentioned in the manuscript (see, l. 363). That step introduced a significant change in R2, which has also been added to the manuscript (see, l. 364). 

If we are correct in understanding 'unpacked' as interpreted, then we appreciate the comment of Reviewer 1 and agree that the interaction should be explained briefly. We have thus added a couple of sentences to explain that for people preferring more right-wing parties the negative effect of higher DRSD on protective behavior was more pronounced (see, l. 371ff). 

While calculating the RAs with mean-centered variables, we noticed a mistake in the computing of our original interaction variable of DRSD and party-preference. We have corrected this mistake, and changed the results accordingly (see, l. 361f). Importantly, the results remain the same, that is, the interaction is still significant. 

13. The description of the goals of Study 2 introduces new construct which were not previously mentioned in the introduction (needs deprivation). I suggest to elaborate on these predictions in separate section in the introduction. In both studies some moderations appeared as a goals of the examinations. However, they were not elaborated and introduce properly in the introduction. Such a situation create a sense of chaotic argumentation and multiple goals which the Authors try to obtained using the one sample

We thank Reviewer 1 for this comment. As indicated in the original manuscript, examining the interaction of needs and DRSD was exploratory in nature. To clarify this, we have added a more extensive explanation of the idea of need deprivation to the introduction of Study 2 (see, l. 406-410).

14. The political orientation measure used in Study 2 should be better described. It consists of 4 Democrats and 4 Republicans, but the calculation is based only on 2 of each category. Please, justify this score calculation. Please also include some data on the validity.

Thanks for this helpful comment. We calculated politician preference score in this way because we think it creates a more differentiated picture of political preference. Because we calculated the score only using the more 'extreme' politicians in their respective party (Ocasio-Cortez and Ilhan Omar for the Democrats; Donald Trump and Majorie Taylor Greene for the Republicans), we can fully cover the political spectrum and not only differentiate between people in the more moderate center. We have also added this more detailed justification in our manuscript (see, l. 490-494). 

15. The other instruments again create a feeling that the Authors include many measures in their study and tried to find some significant associations with the DRSD. It would be better to clearly justify why these instruments were used. I am wondering why the Authors did not controlled for SD in the study 2? Moreover, the study 2 is less associated with the exact wording of the title, which suggest testing the DRSD in the context of the pandemic. Study 2 is more about the associations between the DRSD and political preferences.

We appreciate the concern of Reviewer 1 about the instruments included in Study 2. We strictly followed pre-registration in this Study, or extensively explained differences from pre-registration, which should reduce the Reviewer’s concerns. Study 2 is important, because we consider DRSD as a political and social attitude that should be investigated in the broader political context. The newly added paragraph about DRSD and political orientation (see, l. 105-127) and the slightly reworked paragraph about DRSD and social attitudes (see, l. 141ff) should strengthen this point. In fact, we believe it is important for the validity of our scale to locate it in the broader political context and in association to broader political attitudes. In the ‘pandemic age’, disease-related attitudes are gaining importance (Hartman et al., 2021; Lantz et al., 2022) and thus also influence general political attitudes (see, l. 387-396). 

16. The CFA brought inconsistent results. Both CFI and TLI seem proper (but please provide also criteria of fit which were used), but RMSEA is clearly problematic. Moreover, the Authors mention two-factor solution, which was not previously described. Did the Authors tested the differences between chi square of both one- and two-factor solutions?

We appreciate the comments of Reviewer 1 on the CFA in Study 2. First, criteria of fit were originally based on Hu and Bentler (1999) who suggested cut-off values for TLI and CFI close to .95. This has also been added to the manuscript (see, l. 529ff). Following Kenny et al. (2015), we think that in our specific case a high RMSEA should not be considered a great threat to model fit since RMSEA tends to penalize models with lower df with a positive bias. Considering that our model only had one factor with few variables, we consider our CFA as a model with relatively few df. Therefore, the focus should be more on the proper CFI and TLI values. We have also added information on SRMR values for further confidence in our model (see, l. 539ff). Nonetheless, future research should investigate the factor structure of our scale in more depth. We have added this consideration of the high RMSEA in the manuscript (see, l. 531-534) as well as a part in the limitation section discussing the CFA further (see, l. 717-720). 

Originally the two-factor model was included in the manuscript to show that an alternative model did not show better fit to the data than the hypothesized one factor model. In this two-factor model we had all items directly mentioning Covid-19 load on the one factor and all other items on the other factor. However, in response to the reviewer’s comment we decided to delete this part in the manuscript and instead rely on the justification based on Kenny et al. (2015).

17. The hierarchical RA in Study 2 looks problematic. First, when the goals is to analyze interaction, the Authors have to state how they computed interaction term, and did he model explained more variance when the interaction term was entered. Some standardized betas are higher than 1.00 – please explain such situation. Again, collinearity statistics are necessary.

We appreciate Reviewer 1’s comment on the RA in Study 2. Based on this comment we have added an explanation of how we calculated the interaction term to the manuscript (which was done in the same way as in Study 1; see, l. 573f). Standardized Betas are most likely higher than 1 in some cases because of high multicollinearity. To counteract this, we have added the same RAs with mean centered variables as predictors to the manuscript (see, l. 586-596). We created a new table for the interaction analyses (see Table 7). We have mentioned the explained variance through adding the interaction in a separate step (see, l. 583ff). We also added collinearity statistics to the table for this RA (see Table 6 & 7). 

18. The sections entitled “Is social Darwinism an individual, intergroup, or societal phenomenon?” seem misleading in the discussion. The studies were about the DRSD, not the SD itself. Thus, I suggest to keep to the results of the Authors’ studies. However, the structure of this section could be used to structure the introduction in terms of correlates of the DRSD.

We thank Reviewer 1 for this very helpful observation. We agree and hence we have changed the heading in the Discussion (see, l. 721f). 

All in all, we thank Reviewer 1 for the very helpful ideas and suggestions that really helped stimulate thoughts about our manuscript.

Reviewer 2

1. In my opinion, authors should implement at least one non-self-report measure to check scale’s validity. It is harder to do, but if scales are validated only by other paper-and-pencil measures, we cannot be fully sure what is their relation to real world.

We thank Reviewer 2 for this comment. Study 2 included two more behavioral measures: people's vote in the 2020 US election and a donation question (participants were able to choose one of three organizations to which they would donate $ 100). Using logistical regression analyses, DRSD significantly predicts voting for Donald Trump (vs. Joe Biden) and significantly predicts donating to the National Rifle Association (vs. to the National Association for the Advancement of Colored People), even when controlling for SDO and RWA. We have added these analyses as exploratory analyses to our manuscript to further illustrate the predictive validity of DRSD (see, l. 601-635). And we have mentioned them in the discussion section of Study 2 (see, l. 643f), as well as in the General Discussion (see, l. 705-710). 

Furthermore, as indicated by recent research, we think, that self-reported behavior can serve as a good approximation for actual behavior. Allcott et al. (2020), for example, showed that significant differences in self-reported social distancing behavior between Republicans and Democrats in the US, were also evident in actual social-distancing behavior assessed via smartphone location data. Also, Gollwitzer et al. (2022) showed that self-reported measures of social distancing present valid measures of actual social distancing behavior. In their study, even a short two-item self-report of social distancing (comparable to the one we used in our study) was valid in predicting actual social-distancing assessed using GPS tracking. This gives us additional confidence that our findings are valid and extend to real-world behavior. We have added this line of argumentation to our discussion section (see, l. 738-743). Because we fully share the reviewer’s opinion that actual behavioral measures would provide great validation for our scale, we added this as a direction for future research (see, l. 759-762). 

2. I couldn’t find information about national identity of the participants of the first study; Authors inform in the abstract that they collected data from two nationalities, but it is not evident what is the first one

We thank Reviewer 2 for this observation and comment. We have added the information that Study 1 was conducted in German to the manuscript (see, l. 199, 666). 

3. Beta’s should not be written in italics

We appreciate Reviewer 2’s attention to detail and pointing out this mistake. We have adjusted this throughout our manuscript. 

We also corrected some typos that we realized during the last proofreading process. 

Data, Syntaxes and the Online Supplemental Materials have been updated on the OSF. 

Figure 1 (Hypotheses) 

Figure 2 (Scree-Plot) 

References

Allcott, H., Boxell, L., Conway, J., Gentzkow, M., Thaler, M., & Yang, D. (2020). Polarization and public health: Partisan differences in social distancing during the coronavirus pandemic. Journal of Public Economics, 191. https://doi.org/10.1016/j.jpubeco.2020.104254

Comrey, A. L. (1988). Factor-analytic methods of scale development in personality and clinical psychology. Journal of Consulting and Clinical Psychology, 56(5), 754.

Deutsche Gesellschaft für Psychologie. (2016). Berufsethische Richtlinien des Berufsverbandes Deutscher Psychologinnen und Psychologen e.V. und der Deutschen Gesellschaft für Psychologie e.V.

Gollwitzer, A., McLoughlin, K., Martel, C., Marshall, J., Höhs, J. M., & Bargh, J. A. (2022). Linking self-reported social distancing to real-world behavior during the COVID-19 pandemic. Social Psychological and Personality Science, 13(2), 656–668.

Hartman, T. K., Stocks, T. V. A., McKay, R., Gibson-Miller, J., Levita, L., Martinez, A. P., Mason, L., McBride, O., Murphy, J., & Shevlin, M. (2021). The authoritarian dynamic during the COVID-19 pandemic: Effects on nationalism and anti-immigrant sentiment. Social Psychological and Personality Science, 12(7), 1274–1285.

Hu, L., & Bentler, P. M. (1999). Cutoff criteria for fit indexes in covariance structure analysis: Conventional criteria versus new alternatives. Structural Equation Modeling: A Multidisciplinary Journal, 6(1), 1–55.

Kenny, D. A., Kaniskan, B., & McCoach, D. B. (2015). The performance of RMSEA in models with small degrees of freedom. Sociological Methods & Research, 44(3), 486–507.

Lantz, B., Wenger, M. R., & Mills, J. M. (2022). Fear, Political Legitimization, and Racism: Examining Anti-Asian Xenophobia During the COVID-19 Pandemic. Race and Justice, 21533687221125816.

Lesschaeve, C. (2017). The predictive power of the left-right self-placement scale for the policy positions of voters and parties. West European Politics, 40(2), 357–377.

Rudman, L. A., & Saud, L. H. (2020). Justifying Social Inequalities: The Role of Social Darwinism. Personality and Social Psychology Bulletin, 46(7), 1139–1155. https://doi.org/10.1177/0146167219896924

Schönbrodt, F. D., & Perugini, M. (2013). At what sample size do correlations stabilize? Journal of Research in Personality, 47(5), 609–612.

Tung, H. H., Chang, T.-J., & Lin, M.-J. (2022). Political ideology predicts preventative behaviors and infections amid COVID-19 in democracies. Social Science & Medicine, 308, 115199.

van Assche, J., van Hiel, A., Dhont, K., & Roets, A. (2019). Broadening the individual differences lens on party support and voting behavior: Cynicism and prejudice as relevant attitudes referring to modern‐day political alignments. European Journal of Social Psychology, 49(1), 190–199.

---

## [Decision Letter · Decision Letter 1]

12 Dec 2022

PONE-D-22-19515R1Survival of the Fittest in the Pandemic Age: Introducing Disease-Related Social DarwinismPLOS ONE

Dear Dr. Nachtwey,

Thank you for submitting your manuscript to PLOS ONE. After careful consideration, we feel that it has merit but does not fully meet PLOS ONE’s publication criteria as it currently stands. Therefore, we invite you to submit a revised version of the manuscript that addresses the points raised during the review process. Please submit your revised manuscript by Jan 26 2023 11:59PM. If you will need more time than this to complete your revisions, please reply to this message or contact the journal office at plosone@plos.org. Please include the following items when submitting your revised manuscript:A rebuttal letter that responds to each point raised by the academic editor and reviewer(s). You should upload this letter as a separate file labeled 'Response to Reviewers'.A marked-up copy of your manuscript that highlights changes made to the original version. You should upload this as a separate file labeled 'Revised Manuscript with Track Changes'.An unmarked version of your revised paper without tracked changes. You should upload this as a separate file labeled 'Manuscript'.If applicable, we recommend that you deposit your laboratory protocols in protocols.io to enhance the reproducibility of your results. Protocols.io assigns your protocol its own identifier (DOI) so that it can be cited independently in the future. For instructions see: https://journals.plos.org/plosone/s/submission-guidelines#loc-laboratory-protocols. Additionally, PLOS ONE offers an option for publishing peer-reviewed Lab Protocol articles, which describe protocols hosted on protocols.io. Read more information on sharing protocols at https://plos.org/protocols?utm_medium=editorial-email&utm_source=authorletters&utm_campaign=protocols.

We look forward to receiving your revised manuscript.

Kind regards,

Peter Karl Jonason

Academic Editor

PLOS ONE

Journal Requirements:

Reviewers' comments:

Reviewer's Responses to Questions

**Comments to the Author**

1. If the authors have adequately addressed your comments raised in a previous round of review and you feel that this manuscript is now acceptable for publication, you may indicate that here to bypass the “Comments to the Author” section, enter your conflict of interest statement in the “Confidential to Editor” section, and submit your "Accept" recommendation.

Reviewer #1: All comments have been addressed

Reviewer #2: All comments have been addressed

2. Is the manuscript technically sound, and do the data support the conclusions?

Reviewer #1: Yes

Reviewer #2: Yes

3. Has the statistical analysis been performed appropriately and rigorously? 

Reviewer #1: Yes

Reviewer #2: Yes

4. Have the authors made all data underlying the findings in their manuscript fully available?

Reviewer #1: Yes

Reviewer #2: Yes

5. Is the manuscript presented in an intelligible fashion and written in standard English?

Reviewer #1: Yes

Reviewer #2: Yes

6. Review Comments to the Author

Reviewer #1: The revised version of the manuscript was substantially improved. The Authors put a great effort in the revision and adressed all suggestions of the reviewers. I have now only a few suggestions:

#1. There are some possible problems with collinearity presented in Table 4 (DV: protective behavior; IV: Left/Right; DRSD; Interaction). Similar situation is present in Table 7 (see a lot of Tolerance is below .100). I think that the Authors should at least comment on this possible problem and, preferably, explain it.

#2. Given the study model which indicates that DRSD is a part/form of social darwinism, I think that the Authors could indicate how these two phenomenons are structurally separate. I think that for the clarity, additional EFA on the items of both DRSD and SD could be beneficial. In example, the Authors could include this analysis only as a note.

#3. In Study 1 and Study 2, e.g. a Harman's test for detection of common method bias could be an additional information in favor of the Authors standing that the tested variables were indeed independent and the validity analysis was conducted properly.

Reviewer #2: I would like to congratulate Author(s) on the good work done. In my opinion, manuscript is much better now and deserves for publication as is.

7. PLOS authors have the option to publish the peer review history of their article (what does this mean?). If published, this will include your full peer review and any attached files.

Reviewer #1: **Yes: **Marcin Moroń

Reviewer #2: **Yes: **Jarosław Piotrowski

---

## [Author Response · Author response to Decision Letter 1]

12 Jan 2023

Journal Requirements

We appreciate PLOS ONE’s attention to detail. We have checked our reference list and did not find articles that were retracted or needed to be removed. However, while working on the comments made by Reviewer 1, we found a recently published paper investigating social Darwinism during the COVID-19 pandemic, specifically its relationship to ageism (Kanık et al., 2022). Because we think that this paper is relevant to our line of argumentation, we have added a short paragraph in the introduction citing said paper (l. 94-97). Apart from adding this paper to our reference list, no other changes have been made. 

Reviewer 1

1. The revised version of the manuscript was substantially improved. The Authors put a great effort in the revision and adressed all suggestions of the reviewers. I have now only a few suggestions:

There are some possible problems with collinearity presented in Table 4 (DV: protective behavior; IV: Left/Right; DRSD; Interaction). Similar situation is present in Table 7 (see a lot of Tolerance is below .100). I think that the Authors should at least comment on this possible problem and, preferably, explain it.

First, we want to thank Reviewer 1 for the positive feedback regarding our revisions after the first round of peer-review. Second, we thank Reviewer 1 for bringing up the low Tolerances in some of the regression tables. As can be seen, the low Tolerances in both tables only concern the regression analyses with uncentered predictors. While the tolerances in the regression analyses with centered predictors (marked by © in the tables) are satisfactory. To prevent future confusion, we have added a note to both tables explaining this (see Table 4 &7). 

2. Given the study model which indicates that DRSD is a part/form of social darwinism, I think that the Authors could indicate how these two phenomenons are structurally separate. I think that for the clarity, additional EFA on the items of both DRSD and SD could be beneficial. In example, the Authors could include this analysis only as a note.

We thank Reviewer 1 for his comments on the relationship of DRSD and SD and suggesting an additional EFA. Because we agree with Reviewer 1, we have calculated the proposed EFA and found that this produces two highly correlated (r = .5) though distinct factors; Factor 1 containing all items of DRSD and Factor 2 all items of SD. This provides support for our conception of DRSD as a structurally distinct part of SD. We have added this analysis in a short paragraph to our manuscript (l. 377-382). 

3. In Study 1 and Study 2, e.g. a Harman's test for detection of common method bias could be an additional information in favor of the Authors standing that the tested variables were indeed independent and the validity analysis was conducted properly.

We thank Reviewer 1 for bringing up the concern about common method bias. Even though the Harman’s test has been criticized a lot in the literature (e.g., Aguirre-Urreta & Hu, 2019; Baumgartner et al., 2021), we agree with Reviewer 1 that it could provide additional evidence in our favor. Hence, we have calculated the test for both studies and in both studies the single factor extracted explained around 32 % of total variance. This can be interpreted as additional information giving us confidence in our analyses. We have added a short section describing the test for both Studies (l.392-395 & l. 643-646)

References

Aguirre-Urreta, M. I., & Hu, J. (2019). Detecting common method bias: Performance of the Harman’s single-factor test. ACM SIGMIS Database: The DATABASE for Advances in Information Systems, 50(2), 45–70.

Baumgartner, H., Weijters, B., & Pieters, R. (2021). The biasing effect of common method variance: some clarifications. Journal of the Academy of Marketing Science, 49(2), 221–235.

Kanık, B., Uluğ, Ö. M., Solak, N., & Chayinska, M. (2022). “Let the strongest survive”: Ageism and social Darwinism as barriers to supporting policies to benefit older individuals. Journal of Social Issues, 78(4), 790–814.

---

## [Editor Report · Decision Letter 2]

16 Jan 2023

Survival of the Fittest in the Pandemic Age: Introducing Disease-Related Social Darwinism

PONE-D-22-19515R2

Dear Dr. Nachtwey,

We’re pleased to inform you that your manuscript has been judged scientifically suitable for publication and will be formally accepted for publication once it meets all outstanding technical requirements.

Kind regards,

Peter Karl Jonason

Academic Editor

PLOS ONE
---

## [Editor Report · Acceptance letter]

27 Feb 2023

PONE-D-22-19515R2 

Survival of the Fittest in the Pandemic Age: Introducing Disease-Related Social Darwinism 

Dear Dr. Nachtwey:

I'm pleased to inform you that your manuscript has been deemed suitable for publication in PLOS ONE. Congratulations! Your manuscript is now with our production department. 

Kind regards, 

on behalf of

Dr. Peter Karl Jonason 

Academic Editor

PLOS ONE